# Mapping hematologists' HIV testing behavior among lymphoma patients–A mixed-methods study

Saskia Bogers[1,2,3]*, Hanne Zimmermann[4,5], Amie Ndong[1], Udi Davidovich[5,6], Marie José Kersten[7], Peter Reiss[1,8,9,10], Maarten Schim van der Loeff[1,2,5], Suzanne Geerlings[1,2,3], on behalf of the HIV Transmission Elimination AMsterdam (H-TEAM) Consortium[¶]

1 Amsterdam UMC, Location University of Amsterdam, Internal Medicine, Amsterdam, the Netherlands, 2 Amsterdam Institute for Infection and Immunity, Infectious Diseases, Amsterdam, the Netherlands, 3 Amsterdam Public Health Research Institute, Quality of Care, Amsterdam, the Netherlands, 4 Department of Work and Social Psychology, Maastricht University, Maastricht, the Netherlands, 5 Department of Infectious Diseases, Public Health Service of Amsterdam, Amsterdam, the Netherlands, 6 Department of Social Psychology, University of Amsterdam, Amsterdam, the Netherlands, 7 Amsterdam UMC, Location University of Amsterdam, Hematology, Cancer Center Amsterdam, Amsterdam, the Netherlands, 8 Stichting HIV Monitoring, Amsterdam, the Netherlands, 9 Amsterdam Institute for Global Health and Development, Amsterdam, the Netherlands, 10 Amsterdam UMC, Location University of Amsterdam, Global Health, Amsterdam, the Netherlands

¶ Membership of the HIV Transmission Elimination AMsterdam (H-TEAM) Consortium is provided in the Acknowledgments.
* s.j.bogers@amsterdamumc.nl

**Data Availability Statement:** All relevant data are within the paper and its Supporting Information files.

## Abstract

### Background

HIV testing among patients with malignant lymphoma (PWML) is variably implemented. We evaluated HIV testing among PWML, and mapped factors influencing hematologists' testing behavior.

### Materials

We conducted a mixed-methods study assessing HIV testing among PWML, factors influencing HIV testing and opportunities for improvement in five hospitals in the region of Amsterdam, the Netherlands. The proportion of PWML tested for HIV within 3 months before or after lymphoma diagnosis and percentage positive were assessed from January 2015 through June 2020. Questionnaires on intention, behavior and psychosocial determinants for HIV testing were conducted among hematologists. Through twelve semi-structured interviews among hematologists and authors of hematology guidelines, we further explored influencing factors and opportunities for improvement.

### Findings

Overall, 1,612 PWML were included for analysis, including 976 patients newly diagnosed and 636 patients who were referred or with progressive/relapsed lymphoma. Seventy percent (678/976) of patients newly diagnosed and 54% (343/636) of patients with known lymphoma were tested for HIV. Overall, 7/1,021 (0.7%) PWML tested HIV positive, exceeding

**Funding:** This study was funded by Aidsfonds (grant number: P-42702; funding acquired by SEG. www.aidsfonds.nl) and the HIV Transmission Elimination Amsterdam (H-TEAM) initiative (funding acquired by SEG. www.hteam.nl). The funders had no role in study design, data collection and analysis, decision to publish, or preparation of the manuscript.

**Competing interests:** Dr. Bogers has nothing to disclose. Dr. Zimmermann has nothing to disclose. Dr. Ndong has nothing to disclose. Dr. Davidovich has nothing to disclose. Dr. Kersten reports consulting fees: Kite/Gilead; BMS/Celgene; Novartis; Miltenyi Biotec; Adicet Bio: to institution and payment for honoraria: Kite/Gilead; Roche; BMS/Celgene: to institution and participation on a Data Safety Monitoring Board or Advisory Board: SUBITO study (high dose chemotherapy in breast cancer): No financial compensation. Dr. Reiss reports grants or contracts: Gilead Sciences; ViiV Healthcare; Merck: Investigator-initiated study grants to institution; not related to current work and Participation on a Data Safety Monitoring Board or Advisory Board: Gilead Sciences; ViiV Healthcare; Merck: Honoraria for scientific advisory board participation paid to institution. Dr. Schim van der Loeff has nothing to disclose. Dr. Geerlings has nothing to disclose. This does not alter our adherence to PLOS ONE policies on sharing data and materials.

the 0.1% cost-effectiveness threshold. Questionnaires were completed by 40/77 invited hematologists, and 85% reported intention to test PWML for HIV. In the interviews, hematologists reported varying HIV testing strategies, including testing all PWML or only when lymphoma treatment is required. Recommendations for improved HIV testing included guideline adaptations, providing electronic reminders and monitoring and increasing awareness.

## Conclusions

Missed opportunities for HIV testing among PWML occurred and HIV test strategies varied among hematologists. Efforts to improve HIV testing among PWML should include a combination of approaches.

## Introduction

In 2020, an estimated 2.6 million people were living with HIV in the European region and an estimated 170,000 people became newly infected [1]. Meanwhile, an estimated 33% were unaware of their HIV status, underlining the urgent need for optimal HIV testing [1].

A cost-effective strategy for HIV testing is indicator condition (IC)-guided testing [2–5]. ICs are conditions that are associated with an undiagnosed HIV prevalence of >0.1%, the established cost-effectiveness threshold for HIV screening, that are AIDS-defining, or conditions where failure to identify an HIV infection may have significant adverse implications [6–9]. Even though IC-guided HIV testing is now widely recommended, it is still not a routine practice in the European hospital setting [10–12].

Malignant lymphoma, including both Hodgkin's lymphoma (HL) and non-Hodgkin's lymphoma (NHL), is one of the currently recognized ICs [9]. The risk of developing NHL or HL is markedly increased in people living with HIV (PLHIV) compared to HIV-negative persons [13, 14]. The cumulative incidence of NHL and HL among PLHIV by the age of 75 years is 4.5% and 0.9%, compared to 0.7% and 0.1% among HIV-negative people, respectively [15]. In the Netherlands, NHL is the first occurring AIDS-defining event in 6% of PLHIV, and HL is one of the most common non-AIDS-defining malignancies [16]. HIV testing in patients diagnosed with malignant lymphoma (PWML) may therefore be an important strategy to identify undiagnosed PLHIV, and routine HIV testing at diagnosis is recommended in the guidelines of several lymphoma subtypes [17–19].

We designed a multicenter intervention study (PROTEST 2.0) to assess and subsequently improve IC-guided testing in a selection of ICs, including malignant lymphoma [20, 21]. We found that prior to the intervention, 63% of PWML were tested for HIV within 3 months before or 3 months after lymphoma diagnosis. Stratification by lymphoma type revealed significant variation in HIV testing, with the highest proportions of PWML tested among patients with aggressive types of lymphoma including Burkitt's lymphoma (89%), diffuse large B-cell lymphoma (DLBCL; 73%), mantle cell lymphoma (72%), HL (69%), and T-cell lymphoma (64%), while lowest testing proportions were observed among patients with low-grade types of lymphoma, including lymphoplasmacytic lymphoma (25%), marginal zone or mucosa-associated lymphoid tissue lymphoma (44%), and follicular lymphoma (52%) [21].

As it is unknown which factors influence IC-guided testing for HIV among PWML, in this study, we aimed to assess HIV testing among PWML in more detail, and map factors influencing hematologists' HIV testing behavior among PWML.

## Material and methods

### Study design and setting

This study is part of the PROTEST 2.0 study, which was conducted at two university hospitals, two teaching hospitals and one non-teaching hospital [20]. We performed a mixed-methods study using retrospective data on HIV testing from PWML, and questionnaires and semi-structured interviews among hematologists in the region of Amsterdam, the Netherlands. Data on HIV testing among PWML overall and by subtype have previously been reported in the context of the PROTEST 2.0 study results [21]. Here we report on HIV testing among PWML by diagnosis and treatment status.

### Patient eligibility and assessment of HIV testing

Data from all eligible patients diagnosed with any type of malignant lymphoma in the five participating hospitals from January 2015 through June 2020 were collected. Patients ≥18 years, diagnosed with lymphoma or referred for a second opinion or treatment after lymphoma diagnosis were eligible. Patients without a pathology-confirmed lymphoma diagnosis, those with a known HIV infection prior to lymphoma work-up and diagnosis, and those diagnosed and treated for lymphoma at another hospital were excluded. The primary outcome was the proportion of patients who were tested for HIV within 3 months before or after lymphoma diagnosis. Secondary outcomes were the proportion of patients tested for HIV before initiating lymphoma treatment, the proportion HIV positive, and the proportion of patients who had a CD4 count <350 cells/mm$^3$ at diagnosis (i.e. late-stage HIV infection).

### Questionnaire design and recruitment

Online questionnaires on HIV testing in PWML were conducted anonymously among hematologists in June 2020. All hematology attending physicians and residents in the five hospitals were invited to participate in the questionnaire study by email, and the response ratio was recorded. The questionnaire was based on the Attitude, Social influence and self-Efficacy (ASE) model derived from the Theory of Planned Behavior (TPB) [22, 23]. The final questionnaire contained fifteen 5-point Likert scale questions (1 = most negative response, 5 = most positive response), including one on self-reported HIV testing in PWML, two on intention to test, and twelve on attitudes, norms and self-efficacy regarding HIV testing in PWML (S1 Table).

### Interview design and recruitment

Semi-structured interviews were conducted among hematologists and authors of hematology guidelines in the five hospitals from August 2020 through April 2021. Authors were identified using author lists of currently published national hematology guidelines. The TPB model was used in the design of the interview guide for hematologists [22]. The five domains were knowledge, attitudes, norms, self-efficacy and perceived barriers (S2 Table). The translational research model developed from Rogers' diffusion of innovations model was used in the design of the interview guide for authors of hematology guidelines [24, 25]. The three domains were guideline characteristics (i.e. what is recommended in regards to HIV testing, and who and how is this recommended), communication, and normative systems (S3 Table). All interview participants were additionally requested to suggest opportunities to improve HIV testing in PWML. Convenience sampling was used for participant recruitment. Questionnaire respondents were invited to participate in the interviews at the end of the questionnaire. Additionally, interview participants were recruited through personal invitation by email in all five hospitals, regardless of whether they had completed the questionnaire.

## Data collection

Eligible patients were given the opportunity to opt-out of the use of their data. From the electronic health records (EHR) of included PWML, data on patient demographics, lymphoma diagnosis, and HIV testing were collected using Castor (Castor EDC, Amsterdam, the Netherlands) [20]. Scanned referral letters and other archived documents in patients' EHRs were searched for any evidence of HIV testing done in other settings. LimeSurvey was used for collection of questionnaire data (LimeSurvey GmbH, Hamburg, Germany). As interviews took place during the COVID-19 pandemic, all interviews were conducted through Zoom (Zoom Video Communications Inc., San Jose, California, USA) by AD. The duration of the interviews ranged from 9 to 30 minutes. Audio recordings were made using a secured recording device. No personal identifiers were recorded during the interviews. Interviews were transcribed verbatim by SJB and AD.

## Data analysis

Categorical data collected from EHR of eligible PWML were summarized using frequencies and percentages, and continuous data as means and standard deviations (SD) or medians and interquartile ranges (IQR). Variable distributions were compared between patient groups using unpaired t-tests or Mann-Whitney U tests for continuous data and $X^2$ or Fisher-exact tests for categorical data. Data on questionnaire participant characteristics and factors influencing HIV testing among PWML were summarized descriptively. All quantitative data analyses were performed using Stata 15 (StataCorp LLC, College Station, Texas, USA). AD and SJB coded all transcribed interview data and completed an initial coding system following thematic analysis methods by Braun and Clarke [26]. Subsequently, HZ checked a random sample of 50% for agreement. The final coding system was completed after consensus discussion by SJB and HZ. All qualitative data analyses were performed using MaxQDA 2022 (VERBI Software, Berlin, Germany).

## Ethics statement

All eligible patients were given the opportunity to opt-out of their data being used through written correspondence. All questionnaire and interview participants provided written consent to participation. The Medical Ethics Committee of the Amsterdam UMC, location AMC (METC AMC) determined that this study does not meet the definition of medical research involving human subjects under Dutch law (file no. A1 20.076, 24 February 2020).

## Results

### HIV testing results

A total of 2,961 patient EHRs were screened for eligibility and data of 1,612 patients were included (Fig 1).

Overall, 976 patients (61%) had a new lymphoma diagnosis, and 636 (40%) had a known lymphoma diagnosis but newly entered into care at one of the study sites due to transfer of care, relapsed or progressive disease, or a second opinion. Overall, 1,021 patients (63%) were tested for HIV within 3 months before or after diagnosis [21]; 678/976 (70%) of patients newly diagnosed and 343/636 (54%) of patients who newly entered care. The proportion of patients tested was higher among males than females and higher in younger patients, but did not differ by socio-economic status (Table 1). The proportion of patients tested for HIV varied significantly by hospital (Table 1). By type of lymphoma, significant differences in proportion of patients tested by hospital were observed among patients with Hodgkin's lymphoma

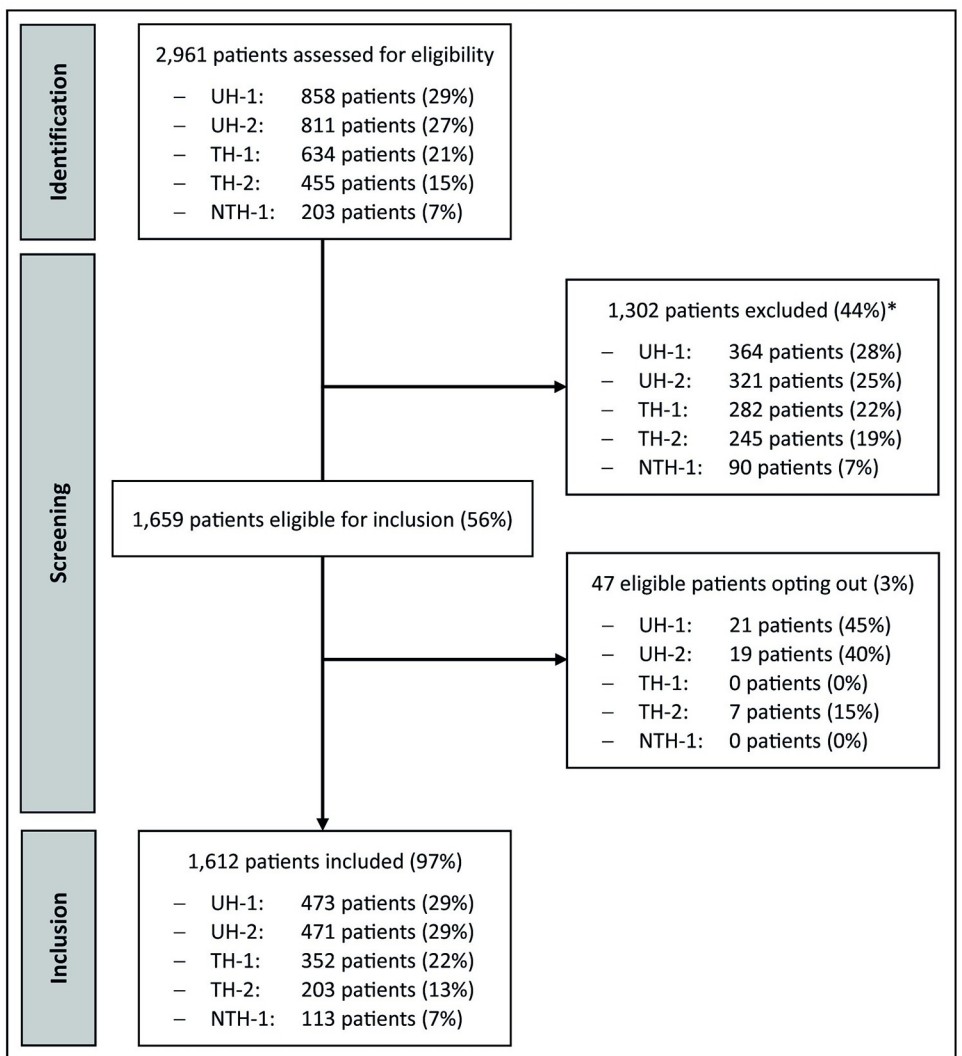

**Fig 1. Flowchart of identification, screening and inclusion of data of patients diagnosed with malignant lymphoma in five hospitals in the region of Amsterdam, 2015–2020.** *Reasons for exclusion were: no definitive lymphoma diagnosis (20%), lymphoma diagnosis and treatment prior completely took place before the study period (57%), diagnosed and treated for lymphoma all took place at another hospital (20%), and known HIV infection prior to lymphoma work-up and diagnosis (3%). NTH: Non-teaching hospital. TH: Teaching hospital. UH: University hospital.

(p = 0.007), DLBCL (p = 0.037), mantle cell lymphoma (p = 0.029), follicular lymphoma (p = 0.001) and marginal zone or mucosa-associated lymphoid tissue lymphoma (p<0.001), but not among patients with T-cell lymphoma (p = 0.364), Burkitt lymphoma (p = 0.891), LPL (p = 0.504), and other NHL types (p = 0.095).

Of 1,306 patients who received treatment for their lymphoma, 928 (71%) were tested for HIV within 3 months before or after diagnosis, of whom 838 (90%) were tested before or on the first day of lymphoma treatment. However, 242/1,306 (19%) of patients who received treatment had no evidence of ever being tested for HIV in their EHR. The remaining 136/1,306 (10%) were tested more than 3 months before or after lymphoma diagnosis. Among patients without any evidence of HIV testing, 139/242 (57%) had a new lymphoma diagnosis requiring immediate treatment.

**Table 1. Characteristics of included patients with malignant lymphoma in five hospitals in the region of Amsterdam, overall and by HIV testing within 3 months before and after malignant lymphoma diagnosis, treatment or work-up, 2015–2020.**

|  | Overall (column %) (n = 1,612) | Tested for HIV (row %) (n = 1,021) | Not tested for HIV (row %) (n = 591) | p value |
|---|---|---|---|---|
| **Sex** |  |  |  | <0.001 |
| Female | 687 (42.6%) | 398 (57.9%) | 289 (42.1%) |  |
| Male | 925 (57.4%) | 623 (67.4%) | 302 (32.7%) |  |
| **Age at lymphoma diagnosis (y)** | 61 (49–71) | 59 (47–69) | 64 (53–74) | <0.001 |
| **Socio-economic status*** |  |  |  | 0.870 |
| Low | 494 (31.0%) | 317 (64.2%) | 177 (35.8%) |  |
| Intermediate | 392 (24.6%) | 250 (63.8%) | 142 (36.2%) |  |
| High | 706 (44.4%) | 443 (62.8%) | 263 (37.3%) |  |
| **Hospital of inclusion** |  |  |  | <0.001 |
| University hospital 1 | 473 (29.3%) | 297 (62.8%) | 176 (37.2%) |  |
| University hospital 2 | 471 (29.2%) | 293 (62.2%) | 178 (37.8%) |  |
| Teaching hospital 1 | 352 (21.8%) | 259 (73.6%) | 93 (26.4%) |  |
| Teaching hospital 2 | 203 (12.6%) | 99 (48.8%) | 104 (51.2%) |  |
| Non-teaching hospital 1 | 113 (7.0%) | 73 (64.6%) | 40 (35.4%) |  |
| **Lymphoma diagnosis** |  |  |  | <0.001 |
| Newly diagnosed at study site | 976 (60.6%) | 678 (69.5%) | 298 (30.5%) | <0.001 |
| Requiring immediate treatment | 832 (85.3%) | 614 (73.8%) | 218 (26.2%) |  |
| Requiring treatment later | 44 (4.5%) | 24 (54.6%) | 20 (45.5%) |  |
| Not requiring treatment | 100 (10.3%) | 40 (40.0%) | 60 (60.0%) |  |
| Known lymphoma diagnosis at presentation | 636 (39.5%) | 343 (53.9%) | 293 (46.1%) | <0.001 |
| Progressive, requiring treatment | 26 (4.1%) | 8 (30.8%) | 18 (69.2%) |  |
| Relapsed lymphoma | 171 (26.9%) | 82 (48.0%) | 89 (52.1%) |  |
| Second opinion | 117 (18.4%) | 20 (17.1%) | 97 (82.9%) |  |
| Transfer from another hospital | 322 (50.6%) | 233 (72.4%) | 89 (27.6%) |  |

Data are depicted as n (%) or median (IQR).

*Twenty patients had a missing socio-economic status.

Of 1,021 patients who were tested for HIV within 3 months before or after diagnosis, seven (0.7%) were HIV positive, all of whom had a CD4 count <350 cells/mm$^3$ at diagnosis (median 97 cells/mm$^3$, IQR 60–130). Of these, five (71%) had DLBCL, one (14%) had Burkitt's lymphoma and one (14%) had T-cell lymphoma. All patients were male and median age at diagnosis was 51 years (IQR 35–57).

## Questionnaire results

The overall response ratio to the hematologists' questionnaire was 40/77 (52%), including 21/40 (53%) in the two university hospitals, 5/21 (24%) in the two teaching hospitals and 14/16 (88%) in the non-teaching hospital. Of respondents, 15 (38%) were attending physicians and 25 (63%) were residents. Median length of work experience in hematology was 4 years (IQR 1–10). Respondents answered all questions. While 27/40 (68%) of respondents reported that they had tested PWML for HIV in the last year, 34/40 (85%) reported intention to test in the future (**Fig 2**). By hospital, self-reported testing in the last year varied significantly (p = 0.032) and was lowest in the non-teaching hospital (mean Likert score 2.8, SD 1.7), followed by the university hospitals (mean 4.3, SD 0.9), while it was highest in the teaching hospitals (mean

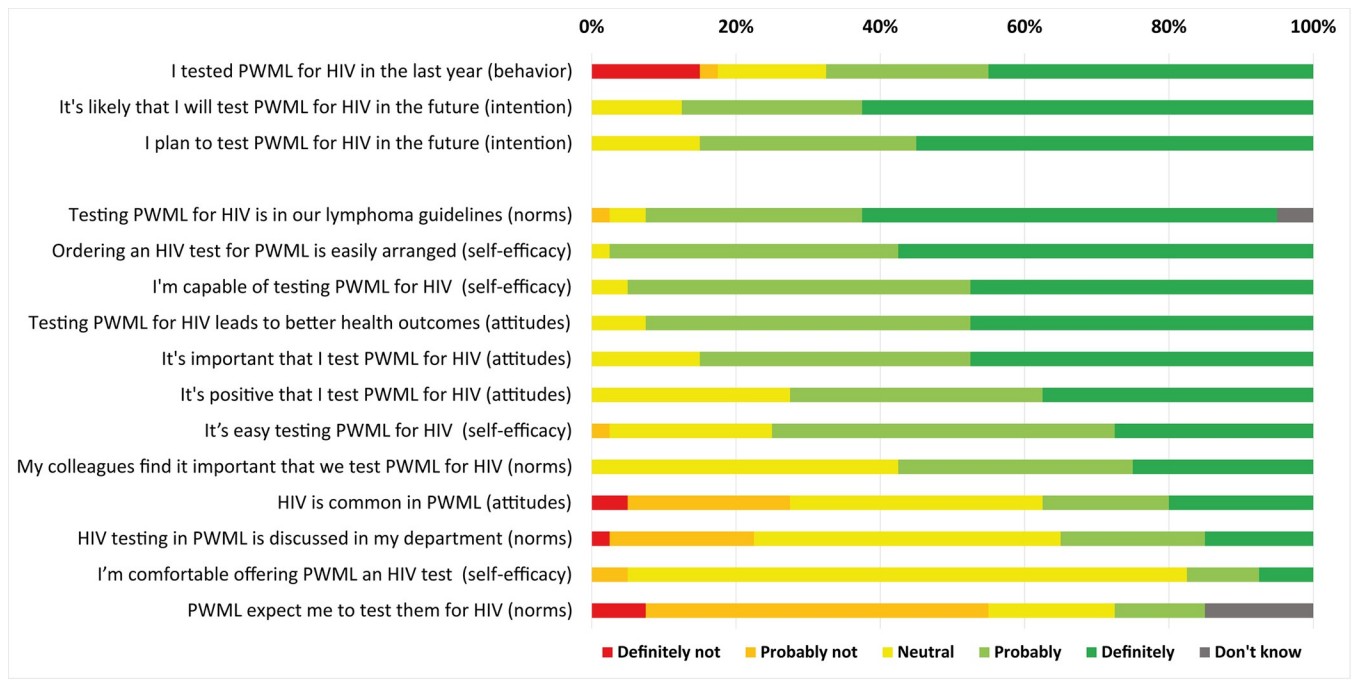

**Fig 2. Questionnaire responses on barriers and facilitators for HIV testing among patients with malignant lymphoma by 40 hematology physicians from 5 hospitals in the region of Amsterdam, 2020, ranked by statement score.** PWML: patients with malignant lymphoma.

4.4, SD 0.9). Intention to test in the future did not differ significantly (mean Likert scores of 4.6, SD = 0.6 in the university hospitals, 4.6, SD = 0.9 in the teaching hospitals, and 4.1, SD = 4.1 in the non-teaching hospital, p = 0.480). Overall, average statement scores were similar across the attitudes, norms and self-efficacy domains. The statements "PWML expect me to test them for HIV" (norms), "HIV testing in PWML is discussed in my department" (norms) and "I'm comfortable offering PWML an HIV test" (self-efficacy) scored lowest on a 5-point Likert scale. The statements "Testing PWML for HIV is in our lymphoma guidelines" (norms), "Ordering an HIV test for PWML is easily arranged" (self-efficacy), and "I'm capable of testing PWML for HIV" (self-efficacy) scored highest (**Fig 2**).

## Interview results

Ten hematologists and two authors of hematology guidelines were interviewed. Median age of respondents was 42 years (IQR 40–46 years). Nine (75%) worked at a university hospital and 3 (25%) at a teaching hospital. No hematologists from the non-teaching hospital could be recruited.

## Interviews with hematologists

From the interviews with the hematologists, seven themes emerged, including **testing strategy**, **timing**, **reasons**, **knowledge**, **norms**, **ways of informing patients**, **patient's responses**, **self-efficacy**, **barriers** and **facilitators** (**Table 2**).

Several HIV **testing strategies** were identified, including testing as part of the diagnostic work-up for lymph node swelling, when recommended by lymphoma guidelines, testing all new lymphoma patients, and only testing when lymphoma treatment was indicated:

> "*It has more to do with the immunosuppressive therapy I am giving them. If they have HIV, I want to know of course.*" (university hospital hematologist, female, 39 years)

**Table 2. Themes identified from semi-structured interviews with hematologists on their HIV testing behavior among lymphoma patients, and influencing factors.**

| |
|---|
| **HIV testing strategy** |
| Test every new lymphoma patient *(n = 8)* |
| Test lymphoma patients only when treating *(n = 5)* |
| Test as part of lymph node swelling/ diagnostic work-up *(n = 3)* |
| Follow the guideline *(n = 2)* |
| **Timing** |
| Test at/before start of treatment *(n = 8)* |
| Test at lymphoma diagnosis *(n = 3)* |
| Timing may vary by lymphoma type *(n = 1)* |
| **Reasons for testing for HIV among lymphoma patients: Clinical relevance for patient and provider** |
| HIV is a risk factor for lymphoma *(n = 6)* |
| HIV negatively influences lymphoma treatment/outcomes *(n = 2)* |
| Important to test to protect staff *(n = 1)* |
| **Knowledge about HIV testing recommendations in guidelines** |
| **Sources of information** |
| National and local guideline *(n = 2)* |
| Educational sessions *(n = 2)* |
| **Type of information** |
| Not sure about the exact guideline recommendation *(n = 7)* |
| Guidelines recommend HIV testing *(n = 5)* |
| Malignant lymphoma is an indicator condition *(n = 2)* |
| Don't use of guideline *(n = 1)* |
| **Perception of guidelines** |
| It feels safe to have HIV testing recommended in guidelines *(n = 1)* |
| **Norms on testing for HIV among lymphoma patients** |
| **Colleagues** |
| Colleagues' mentioning HIV testing reminds me *(n = 4)* |
| Disagree with colleagues not routinely testing for HIV *(n = 1)* |
| I bring it up/test more to remind others *(n = 1)* |
| Not influenced by colleagues *(n = 1)* |
| **Patients** |
| Not influenced if patients have low perceived risk *(n = 1)* |
| Not influenced if patients are afraid *(n = 1)* |
| **Ways of informing patients** |
| Inform it's part of routine work-up *(n = 8)* |
| Do not ask permission, only inform *(n = 6)* |
| Inform on clinical relevance for treatment plan/outcomes *(n = 5)* |
| Ask for permission *(n = 2)* |
| It's more effort to inform patients not in the risk category *(n = 2)* |
| Stress confidentiality *(n = 1)* |
| **Lymphoma patient responses to HIV test offering** |
| Patients rarely/never do not agree *(n = 9)* |
| Patients perceived discrimination because of sexual orientation *(n = 2)* |
| Surprised response by patients *(n = 2)* |
| Patients need for more information before agreeing *(n = 1)* |
| **Hematologists' perceived self-efficacy to test for HIV** |
| Good self-perceived efficacy *(n = 4)* |

*(Continued)*

**Table 2.** (Continued)

| |
|---|
| **Good perceived self-efficacy to deliver positive test result** |
| Positive tests are rare *(n = 3)* |
| Like telling them they have lymphoma/also bad news *(n = 2)* |
| I am able to do that *(n = 2)* |
| Low perceived self-efficacy in counseling/referral for counsel *(n = 5)* |
| **Barriers to HIV testing** |
| **Provider level—related to patients' characteristics** |
| Done less in older patients *(n = 3)* |
| When patients are accompanied *(n = 2)* |
| Anticipated patient feeling stigmatized *(n = 1)* |
| **Provider level—general** |
| Testing might have been done in referring hospital *(n = 5)* |
| Forget if you didn't do it right away *(n = 1)* |
| Don't re-test in recurrent lymphoma *(n = 1)* |
| **System level** |
| Lack of awareness on guideline *(n = 2)* |
| Not in standard order set *(n = 1)* |
| IC guided testing not implemented well enough in general *(n = 1)* |
| **Contextual level** |
| Not a clear association of HIV with all lymphoma types *(n = 5)* |
| HIV is not common *(n = 1)* |
| **Facilitators for HIV testing** |
| **Working environment** |
| Low threshold in Amsterdam because it's more common *(n = 3)* |
| Training/working in HIV treatment center, more routine *(n = 2)* |
| **It is routine/part of the work-up** |
| Because it is included in the routine test/order set *(n = 4)* |
| Because it's in the guidelines *(n = 1)* |
| No longer a big taboo like in the past *(n = 2)* |
| Readily available *(n = 1)* |
| If a patient asks for an HIV test it is easy *(n = 1)* |
| You no longer have to ask for elaborate informed consent *(n = 1)* |

Accordingly, the **timing of testing** varied among respondents, with some reporting to test immediately at lymphoma diagnosis, while others test before/at the start of lymphoma treatment. One hematologist reported that timing of testing may vary:

"*The timing of the testing may be different because in patients with chronic lymphocytic leukemia for example I do it at diagnosis but some people only do it when they start the treatment.*" (university hospital hematologist, female, 42 years)

**Reasons for HIV testing** among PWML included clinical relevance for patients and provider: HIV being a risk factor for lymphoma, HIV negatively affecting the lymphoma treatment outcome, and to protect staff from occupational infection. Hematologists reported that they gained their **knowledge** on HIV testing recommendations from national and local guidelines, as well as educational sessions. However, the majority (n = 7) reported not knowing the exact guideline recommendations on HIV testing, while others (n = 5) reported that the guidelines do recommend HIV testing. Regarding **norms**, several hematologists reported that they

were reminded of HIV testing when colleagues mentioned it, while one reported routinely mentioning HIV testing to remind others. No hematologists reported that patients' beliefs or attitudes regarding HIV testing influenced their own HIV testing behavior.

On **ways of informing patients**, the majority (n = 8) of hematologists reported that they inform the patient that it is part of a routine workup for lymphoma, and six hematologists added that they do not explicitly ask for permission to test for HIV, while two reported explicitly asking for permission. Five reported they inform the patient on the clinical relevance of HIV testing for the treatment and outcome of their lymphoma, and two mentioned that they found informing their patient of HIV testing to require more effort when patients did not belong to HIV key groups:

> "*If you have a 23 year old student with a lymphoma, I will explain that to them that we test for viral associations. It's something that we mention, but it doesn't change our practice. But it takes more effort to explain why you're testing that.*" (teaching hospital hematologist, male, 42 years)

On **patients' responses** to being offered an HIV test, nearly all respondents (n = 9) reported that patients rarely or never refuse an HIV test, although some reported that patients may sometimes respond surprised, need more information before agreeing, or perceive discrimination when offered HIV testing because of their sexual orientation. No hematologists reported that they have low perceived **self-efficacy** to test for HIV. However, while two reported that they are well-trained at delivering news of a positive HIV test result as it is just like informing patients of a cancer diagnosis, five reported low perceived self-efficacy in counselling patients with newly diagnosed HIV, and will refer them to an HIV specialist for this.

Contextual level, provider level, and system level **barriers** to HIV testing among PWML were identified (**Table 2**). Contextual level barriers included HIV not being common, and a lack of a clear association between HIV and incidence of some types of lymphoma:

> "*For Hodgkin lymphoma I understand that you do not test it immediately because the association with HIV is not that clear as with DLBCL.*" (university hospital hematologist, female, 39 years)

General provider level barriers included testing having been done in the referring hospital, and forgetting to test when it's not done right away:

> "*Sometimes you think we've already performed a test, because in a lot of patients when I see them first, I already do the HIV testing. And it could be that sometimes you forget, and then later on, you think you've done it already, and you don't look back.*" (teaching hospital hematologist, female, 41 years)

Provider level barriers specifically related to patients' characteristics included older age, where hematologist's mentioned low perceived risk in older patients, anticipating patients feeling stigmatized, and patients being accompanied:

> "*I think it's easier to talk about it when people are not sitting here with their whole family. That's a barrier. Also when it's a heterosexual relationship, I might feel that it's more difficult to talk about it than if it's a homosexual relationship.*" (teaching hospital hematologist, female, 41 years)

Identified **facilitators** for HIV testing included it no longer being a big taboo and informed consent no longer being required, making it easier and less time-consuming to discuss, and HIV testing being part of a routine or guideline and readily available:

"*It's a package with hepatitis B and C, everything is in it. You don't have to think about it.*" (university hospital hematologist, female, 39 years)

"*The test is easy, available and just part of the workup. So, I don't have any barriers.*" (university hospital hematologist, male, 47 years)

Additionally, HIV being relatively common in Amsterdam, as well as working in an HIV treatment center where HIV testing is more routinely done were identified facilitators for HIV testing.

"*I worked in [HIV treatment center] for a while where I learned to do the testing, because there they talk easily about it because they have so many patients with HIV that it's a normal thing.*" (university hospital hematologist, female, 42 years)

## Interviews with authors of hematology guidelines

From the interviews with authors of hematology guidelines, themes on the reason and evidence for HIV testing recommendations in lymphoma guidelines, and themes on guideline development and communication on guidelines with end-users emerged. The guideline authors mentioned that HIV testing recommendations are included in lymphoma guidelines because an HIV infection would influence the treatment plan, because lymphoma is more common among PLHIV and might be a presenting symptom of HIV, and because it is important to have a uniform HIV testing recommendation for PWML. On the evidence for HIV testing among PWML, respondents stated that there is no extensively researched recommendation for HIV testing in the lymphoma guidelines, because they thought it is already common practice and not a matter of debate when developing the guidelines. However, one respondent clarified that the evidence for HIV testing likely varies by lymphoma type. On guideline development and communication, the authors mentioned that the target audience for the guidelines, i.e. hematologists, are actively invited by email to respond to concept guidelines for approval and feedback, and updated guidelines are disseminated and presented at hematology meetings, in the Dutch hematology journal, and through the Dutch hematology society's website and electronic mailings, assuring that all involved may give feedback and are informed on guidelines. However, one respondent added that local implementation of guideline recommendations including routine HIV testing in PWML is not monitored.

## Reflections and recommendations on observed proportions of patients HIV tested

All twelve interview participants reflected on the measured proportions of patients HIV tested among PWML and were asked for recommendations for improvement. Most hematologists had expected higher proportions of patients tested based on their own HIV testing behavior and acknowledged a need for improvement (**Table 3**). Ten respondents recommended increasing awareness among hematologists on the need for HIV testing among PWML. Examples for increased awareness included electronic solutions such as prompts in the EHR of PWML, presenting HIV testing recommendations at conferences or in hematology journals

**Table 3. Hematologists' reflections on HIV testing among patients with malignant lymphoma and recommendations for improvement.**

| Theme | Example |
|---|---|
| **Expected/should be higher (at my hospital) (n = 10)** | |
| Expected higher proportions of patients HIV tested (n = 5) | "I didn't really believe it, I said I test all my patients, and we actually checked if it weren't my patients but we were actually kind of shocked by the result." (teaching hospital hematologist, female, 41) |
| If you don't test for HIV you should have a good reason (n = 1) | "If there is a very good reason not to do it, I mean if somebody is already known HIV positive we don't need to do it again but if you don't know the HIV status, at least you should write down in your file why you don't do the test." (university hospital hematologist, male, 47) |
| It's a missed opportunity not to test for HIV (n = 1) | "I think it's a missed opportunity. We should educate our people better." (university hospital hematologist, male, 47) |
| **We need more awareness on HIV testing recommendations (n = 10)** | |
| We need a system to remind us of HIV testing (n = 5) | "We could be helped by the system. So the computer system could for example pop up and say well, you entered a new diagnosis of lymphoma, did you do an HIV test, do you want to do it right now? That could be really good." (university hospital hematologist, male, 39) |
| Hematologists should be reminded of the guidelines (n = 3) | "I think because the guidelines are quite clear I think some people should be reminded this is part of the routine testing specially certain types of lymphoma." (university hospital hematologist, female, 37) |
| Present HIV testing recommendation and evidence at conference/in national hematology journal (n = 3) | "I think it would help to put this issue with lack of HIV testing in the Dutch hematology conference or something like that. I think that would help." (teaching hospital hematologist, female, 40) |
| Audit and feedback HIV testing implementation (n = 2) | "I think it's just showing the numbers and making people see the importance of it I think by giving a presentation, making people aware of these numbers I think it can help." (university hospital hematologist, female, 39) |
| Mention HIV testing every day when there are new patients (n = 2) | "We have these meetings where we discuss new lymphoma patients. One of us should be asking always what about HIV." (university hospital hematologist, female, 58) |
| **Add HIV testing to all lymphoma guidelines (n = 4)** | "I think it would be most feasible as an implementation strategy if you basically say lymphoma equals HIV test." (university hospital hematologist, male, 39) |
| **Would be good to check if we missed any HIV diagnoses (n = 1)** | "It would be good to do an HIV test in these patients who didn't receive an HIV test to see if it's the case if we've missed any HIV patient." (university hospital hematologist, female, 39) |
| **Likely no need to improve it in all lymphoma types (n = 1)** | "If it's definitely not indicated, if you're 70+ and you're diagnosed with a low-grade lymphoma, and in the past 5 years, nobody with those characteristics has ever been tested positive for HIV, then you might also consider not testing that group, you know." (teaching hospital hematologist, male, 42) |

and receiving audit and feedback on HIV testing. Additionally, some respondents recommended adding HIV testing recommendations to all lymphoma guidelines and checking whether any opportunities for HIV diagnosis were previously missed among PWML. Conversely, one respondent pointed out that HIV testing might not need improving in all PWML patients due to low HIV prevalence, such as among elderly patients with low-grade lymphoma (**Table 3**).

## Discussion

We assessed HIV testing by hematologists among PWML and mapped factors influencing hematologists' testing behavior. Overall, 63% of all PWML [21], and 70% of newly diagnosed PWML were tested for HIV within 3 months before or after lymphoma diagnosis. While 71% of patients who received treatment for lymphoma had been tested for HIV within 3 months, 10% was tested less recently, and 19% had no evidence of ever having been tested, revealing opportunities for improvement. The observed HIV positivity percentage of 0.7% among PWML exceeded the previously established cost-effectiveness threshold for HIV screening of 0.1% [9]. However, as newly diagnosed patients had either DLBCL, Burkitt lymphoma, or T-cell lymphoma, cost-effectiveness of this testing strategy among other types of malignant lymphoma could not be verified in this study.

In the questionnaires and interviews, hematologists reported that their intention to test for HIV among PWML is high, although some reported that this varies by lymphoma type, patient characteristics such as age, and whether lymphoma treatment is required. This is reflected in our observed proportions of patients tested for HIV, which were highest among patients diagnosed with types of lymphoma requiring immediate treatment, and among younger male patients. We observed that there is disagreement among hematologists as to whether patients with all types of lymphoma should be tested for HIV or not, and guideline authors mentioned that the evidence supporting HIV testing may vary by lymphoma type.

Research showed that in the era of effective antiretroviral therapy for HIV, the risk of NHL is elevated 11-fold among PLHIV compared to the general population, but this risk varies substantially by type of NHL, and is not increased among some types, with a standardized incidence ratio of 1.0 (95% CI 0.4–2.3) for mantle cell lymphoma, and 0.8 (95% CI 0.5–1.2) for chronic lymphocytic leukemia/small lymphocytic lymphoma [21]. Not routinely testing for HIV among some types of (indolent) lymphoma may therefore be justified. However, while only 25% of patients with lymphoplasmacytic lymphoma (LPL) in the PROTEST 2.0 study was tested for HIV within 3 months [21], the previously identified standardized incidence ratio of LPL among PLHIV was 3.6 (95% CI 2.0–6.0) [27], highlighting likely missed opportunities for HIV diagnosis among patients with low-grade NHL subtypes.

Several hematologists stated that they do not routinely test in patients not requiring treatment for their lymphoma, and the observed proportions HIV tested among patients not requiring treatment was 40%, versus 74% among patients requiring immediate treatment. A possible explanation is that hematologists may not have reduction of undiagnosed HIV as their primary goal, but are more focused on mitigating adverse outcomes of lymphoma treatment, potentially leading to missed opportunities for HIV diagnosis. Additionally, the proportion tested among patients who required treatment later on due to progressive disease was only 31%, indicating missed opportunities for HIV testing in cases where testing is not performed among patients not requiring (immediate) treatment. This is reflected in the interviews, during which hematologists mentioned they do not re-test for HIV in relapsed lymphoma and may forget to test when it is not done right away.

We identified several additional factors influencing hematologists' routine HIV testing behavior among PWML. Provider level barriers related to patients' characteristics included older age, discomfort discussing HIV testing when patients are accompanied, and anticipation of patients feeling stigmatized. These findings highlight the importance of communication skills when offering an HIV test, as well as education on HIV epidemiology among hematologists, as 26% of individuals newly diagnosed with HIV in 2020 were 50 years or older [16].

Identified facilitators for HIV testing included adding routine HIV testing to guidelines and standard laboratory order sets, while recommendations for improvement included

implementing electronic prompts in EHRs. Such prompts may be especially helpful to prevent missed testing opportunities in cases where testing was not done right away. Previous research showed that electronic prompts may increase HIV testing, although their effect is often lost when deactivated [28]. Additionally, guideline recommendations are often not adhered to without additional efforts to increase awareness of, and agreement on such recommendations [29–31]. Likewise, in our study hematologists recommended increasing awareness of HIV testing recommendations and guidelines through presentations at national conferences or in hematology journals, educational sessions, discussion of HIV testing during patient review meetings and audit and feedback. These examples all highlight that making HIV testing a routine procedure, and communicating it as such with patients, helps implement HIV testing practices. A combination approach of guideline adaptations, electronic prompt systems, educational interventions and routine reminders is likely to be most effective in improving routine HIV testing among PWML.

## Strengths and limitations

A strength of this study is the mixed-methods approach used to evaluate both actual HIV testing, and factors influencing hematologists' HIV testing behavior among PWML. To our knowledge, this is the first study aiming to understand hematologists' HIV testing behavior. Including questionnaires and interviews with hematologists provided deeper insight into determinants for HIV testing, in addition to assessing proportions of patients who were tested for HIV. The considerable range in duration of interviews may suggest that the interviewed participants formed a heterogeneous group in terms of subject engagement. However, some limitations should be noted. While we attained a relatively high response ratio to the questionnaire, only a small number of hematologists, and none from the non-teaching hospital could be recruited for an interview, possibly leading to response bias and limited generalizability. Additionally, we did not include the patient's perspective in our study. Interviewing this stakeholder group, or conducting focus groups involving all stakeholders, may have led to additional insights in factors influencing HIV testing as well as opportunities for improvement. Finally, as some of the findings in this study are specific to the Dutch setting, generalization of our findings to other settings should only be done with caution.

## Conclusions

This study provided insight into the implementation of routine HIV testing among PWML and factors influencing hematologists' testing behavior. HIV testing was done in a relatively small majority of PWML. Missed opportunities for testing occurred, likely due to lack of HIV testing recommendations in guidelines for some lymphoma types, and conflicting testing strategies among hematologists. The overall HIV positivity percentage confirmed the cost-effectiveness of routine testing among PWML. Efforts to improve its implementation should entail a combination of approaches, including increasing awareness and fortifying rationales for testing, guideline adaptations, providing electronic reminders and monitoring, and increasing institutional and normative support for this testing strategy.

## Supporting information

**S1 Table. Content of online questionnaire on barriers and facilitators for HIV testing among patients with malignant lymphoma.**
(DOCX)

**S2 Table. Interview guide for semi-structured interviews with hematologists working in the region of Amsterdam on factors influencing HIV testing behavior among malignant lymphoma patients.**
(DOCX)

**S3 Table. Interview guide for semi-structured interviews with authors of hematology guidelines working in the region of Amsterdam on the extent of and reasons for HIV testing recommendations in malignant lymphoma guidelines.**
(DOCX)

## Acknowledgments

The authors thank all hematologists who contributed to this study: C. Alhan, M. Chamuleau, M. Donker, J. Heijmans, A. Kater, M. Kersten, I. Kuipers, H. Mooij, S. Tonino, J. Vos, B. van Zaanen, and J. Zijlstra. We thank all coordinating physicians from the study hospitals Amsterdam UMC location AMC, Amsterdam UMC location VUmc, BovenIJ ziekenhuis Amsterdam, Flevoziekenhuis Almere and Onze Lieve Vrouwe Gasthuis Amsterdam: prof. dr. S.E. Geerlings, dr. K. Sigaloff, drs. N. Bokhizzou, dr. J. Branger and prof. dr. K. Brinkman. We further acknowledge all members of the HIV Transmission Elimination AMsterdam (H-TEAM) Consortium.

HIV Transmission Elimination AMsterdam (H-TEAM) Consortium:

Lead author:

G.J. de Bree (g.j.debree@amsterdamumc.nl).

H-TEAM members:

T. van Benthem[1], D. Bons[2], G.J. de Bree[3;4], P. Brokx[5], U. Davidovich[1;6], F. Deug[7], S.E. Geerlings[4], M. Heidenrijk[3], E. Hoornenborg[1], M. Prins[1;4], P. Reiss[3;5], A. van Sighem[8], M. van der Valk[4;8], J. de Wit[9], W. Zuilhof[7].

H-TEAM Project Management:

N. Schat[3], D. Smith[3].

H-TEAM additional collaborators:

M. van Agtmael[10], J. Ananworanich[11], D. Van de Beek[12], G.E.L. van den Berk[13], D. Bezemer[8], A. van Bijnen[7], J.P. Bil[1], W.L. Blok[12], S.J. Bogers[4], M. Bomers[10], A. Boyd[1;8], W. Brokking[14], D. Burger[15], K. Brinkman[13], N. Brinkman[13], M. de Bruin[16], S. Bruisten[1], L. Coyer[1], R. van Crevel[17], M. Dijkstra[1], Y.T. van Duijnhoven[1], A. van Eeden[14], L. Elsenburg[14], M.A.M. van den Elshout[1], E. Ersan[18], P. E. V. Felipa[1], T.B.H. Geijtenbeek[19], J. van Gool[1], A. Goorhuis[4], M. Groot[14], C.A. Hankins[3], A. Heijnen[20;21], M.M.J Hillebregt[8], M. Hommenga[1], J.W. Hovius[4], Y. Janssen[22], K. de Jong[1], V. Jongen[1], N.A. Kootstra[23], R.A. Koup[24], F.P. Kroon[25], T.J.W. van de Laar[26;27], F. Lauw[28], M.M. van Leeuwen[5], K. Lettinga[29], I. Linde[1], D.S.E. Loomans[1], I.M. van der Lubben[1], J.T. van der Meer[4], T. Mouhebati[7], B.J. Mulder[1], J. Mulder[30], F.J. Nellen[4], A. Nijsters[7], H. Nobel[4], E.L.M. Op de Coul[31], E. Peters[10], I.S. Peters[1], T. van der Poll[4], O. Ratmann[32], C. Rokx[33], M.F. Schim van der Loeff[1;34], W.E.M. Schouten[13], J. Schouten[1], J. Veenstra[29], A. Verbon[33], F. Verdult[5], J. de Vocht[10], H.J. de Vries[1;34;35], S. Vrouenraets[29], M. van Vugt[4], W.J. Wiersinga[4], F.W. Wit[4;6], L.R. Woittiez[4], S. Zaheri[8], P. Zantkuijl[7], A. Żakowicz[36], M.C. van Zelm[37], H.M.L. Zimmermann[1].

Affiliations:

1. Department of Infectious Diseases, Public Health Service of Amsterdam, Amsterdam, the Netherlands

2. Trans United Europe, Amsterdam, The Netherlands

3. Department of Global Health, Amsterdam UMC–location AMC, and Amsterdam Institute for Global Health and Development, Amsterdam, the Netherlands

4. Department of Internal Medicine, Division of Infectious Diseases, Amsterdam UMC–location AMC, Amsterdam, the Netherlands

5. Dutch Association of PLHIV, Amsterdam, the Netherlands

6. Department of Social Psychology, University of Amsterdam, Amsterdam, the Netherlands

7. Soa Aids Nederland, Amsterdam, the Netherlands

8. Stichting HIV Monitoring, Amsterdam, the Netherlands

9. Department of Interdisciplinary Social Science: Public Health, Utrecht University, Utrecht, the Netherlands

10. Department of Internal Medicine, Amsterdam UMC–location VUMC, Amsterdam, the Netherlands

11. US Military HIV Research Program and the Henry M. Jackson Foundation for the Advancement of Military Medicine, Bethesda, United States

12. Center of Infection and Immunity Amsterdam (CINIMA), Department of Neurology, Amsterdam UMC–location AMC, Amsterdam, the Netherlands

13. Department of internal medicine, OLVG–location East, Amsterdam, the Netherlands

14. DC Klinieken, Amsterdam, the Netherlands

15. Department of Pharmacy, Radboud University Nijmegen Medical Center, Nijmegen, the Netherlands

16. Aberdeen Health Psychology Group, Institute of Applied Health Sciences, University of Aberdeen, Aberdeen, United Kingdom

17. Department of Internal Medicine, Radboud University Nijmegen Medical Center, Nijmegen, the Netherlands

18. Department of General Practice, Amsterdam UMC–location AMC, University of Amsterdam, Amsterdam, the Netherlands

19. Laboratory of Experimental Immunology, Amsterdam UMC–location AMC Amsterdam, the Netherlands

20. Sexology Center Amsterdam, Amsterdam, the Netherlands

21. GP practice Heijnen & de Meij, Amsterdam, the Netherlands

22. Primary Care Amsterdam and Almere (Elaa), Amsterdam, the Netherlands

23. Laboratory for Viral Immune Pathogenesis, Amsterdam UMC–location AMC Amsterdam, the Netherlands

24. Immunology Laboratory, Vaccine Research Center, National Institute of Allergy and Infectious Diseases, National Institutes of Health, Rockville, Maryland, USA

25. Department of Infectious Diseases, Leiden University Medical Center, Leiden, the Netherlands

26. Department of Medical Microbiology, OLVG, Amsterdam, the Netherlands

27. Department of Donor Medicine Research, Laboratory of Blood-borne Infections, Sanquin Research, Amsterdam, the Netherlands

28. Department of Internal Medicine, Medical Center Jan van Goyen, Amsterdam, the Netherlands

29. Department of Internal Medicine, OLVG–location West, Amsterdam, the Netherlands

30. Department of Internal Medicine, Slotervaart Hospital (former), Amsterdam, the Netherlands

31. Epidemiology and Surveillance Unit, Center for Infectious Disease Control, National Institute of Public Health and the Environment, the Netherlands

32. School of Public Health, Faculty of Medicine, Imperial College London, London, United Kingdom

33. Department of Internal Medicine and Infectious Diseases, Erasmus Medical Center, Rotterdam, the Netherlands

34. Center for Infection and Immunology, Amsterdam (CINIMA), Amsterdam UMC–location AMC, University of Amsterdam, Amsterdam, the Netherlands

35. Department of Dermatology, Amsterdam UMC–location AMC, University of Amsterdam, Amsterdam, the Netherlands

36. AIDS Healthcare Foundation, Amsterdam, the Netherlands

37. Department of Virology, Erasmus Medical Center, Rotterdam, the Netherlands

## Author Contributions

**Conceptualization:** Saskia Bogers, Amie Ndong, Udi Davidovich, Peter Reiss, Suzanne Geerlings.

**Data curation:** Saskia Bogers, Hanne Zimmermann, Amie Ndong.

**Formal analysis:** Saskia Bogers, Hanne Zimmermann, Amie Ndong.

**Funding acquisition:** Suzanne Geerlings.

**Investigation:** Saskia Bogers, Hanne Zimmermann, Amie Ndong, Marie José Kersten.

**Methodology:** Saskia Bogers, Hanne Zimmermann, Amie Ndong, Udi Davidovich, Peter Reiss, Maarten Schim van der Loeff, Suzanne Geerlings.

**Project administration:** Saskia Bogers.

**Resources:** Suzanne Geerlings.

**Supervision:** Udi Davidovich, Marie José Kersten, Peter Reiss, Maarten Schim van der Loeff, Suzanne Geerlings.

**Validation:** Marie José Kersten, Peter Reiss.

**Writing – original draft:** Saskia Bogers.

**Writing – review & editing:** Saskia Bogers, Hanne Zimmermann, Amie Ndong, Udi Davidovich, Marie José Kersten, Peter Reiss, Maarten Schim van der Loeff, Suzanne Geerlings.

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
