## [Decision Letter · Decision Letter 0]

28 Nov 2022

PONE-D-22-26866Mapping hematologists’ HIV testing behavior among lymphoma patients – A mixed-methods studyPLOS ONE

Dear Dr. Bogers,

Thank you for submitting your manuscript to PLOS ONE. After careful consideration, we feel that it has merit but does not fully meet PLOS ONE’s publication criteria as it currently stands. Therefore, we invite you to submit a revised version of the manuscript that addresses the points raised during the review process. Please submit your revised manuscript by Jan 12 2023 11:59PM. If you will need more time than this to complete your revisions, please reply to this message or contact the journal office at plosone@plos.org. Please include the following items when submitting your revised manuscript:A rebuttal letter that responds to each point raised by the academic editor and reviewer(s). You should upload this letter as a separate file labeled 'Response to Reviewers'.A marked-up copy of your manuscript that highlights changes made to the original version. You should upload this as a separate file labeled 'Revised Manuscript with Track Changes'.An unmarked version of your revised paper without tracked changes. You should upload this as a separate file labeled 'Manuscript'.If applicable, we recommend that you deposit your laboratory protocols in protocols.io to enhance the reproducibility of your results. Protocols.io assigns your protocol its own identifier (DOI) so that it can be cited independently in the future. For instructions see: https://journals.plos.org/plosone/s/submission-guidelines#loc-laboratory-protocols. Additionally, PLOS ONE offers an option for publishing peer-reviewed Lab Protocol articles, which describe protocols hosted on protocols.io. Read more information on sharing protocols at https://plos.org/protocols?utm_medium=editorial-email&utm_source=authorletters&utm_campaign=protocols.

We look forward to receiving your revised manuscript.

Kind regards,

Simon White

Academic Editor

PLOS ONE

Journal Requirements:

2. Please include your ethics statement in the Methods section of your manuscript. In the Methods section of your revised manuscript, please include the full name of the institutional review board or ethics committee that granted ethical approval or exempt the study from ethical approval, the approval or permit number that was issued, and the date that approval/exemption was granted.

    "Dr. Bogers has nothing to disclose. Dr. Zimmermann has nothing to disclose. Dr. Ndong has nothing to disclose. Dr. Davidovich has nothing to disclose. Dr. Kersten reports consulting fees: Kite/Gilead; BMS/Celgene; Novartis; Miltenyi Biotec; Adicet Bio: to institution and payment for honoraria: Kite/Gilead; Roche; BMS/Celgene: to institution and participation on a Data Safety Monitoring Board or Advisory Board: SUBITO study (high dose chemotherapy in breast cancer): No financial compensation. Dr. Reiss reports grants or contracts: Gilead Sciences; ViiV Healthcare; Merck: Investigator-initiated study grants to institution; not related to current work and Participation on a Data Safety Monitoring Board or Advisory Board: Gilead Sciences; ViiV Healthcare; Merck: Honoraria for scientific advisory board participation paid to institution. Dr. Schim van der Loeff has nothing to disclose. Dr. Geerlings has nothing to disclose."

5. One of the noted authors is a group or consortium "HIV Transmission Elimination AMsterdam (H-TEAM) Consortium". In addition to naming the author group, please list the individual authors and affiliations within this group in the acknowledgments section of your manuscript. Please also indicate clearly a lead author for this group along with a contact email address.

Additional Editor Comments:

Please address the following issues, in addition to responding to the reviewers comments below:A completed COREQ checklist is required, or other relevant checklists listed by the Equator Network, such as the SRQR, to ensure complete reportingIn the introduction, the second sentence should be changed to read, "…33% were aware…"The statement in the first sentence of the third paragraph needs referencingClarify whether there was any piloting of the questionnaire before useClarify how haematology guideline authors were identified to invite them to participate in interviews?In the discussion there are two sentences that need grammatical amendments. My suggestions are to change them to read, “...this is reflected in our observed proportions of people with HIV tested…” and, “…highest among people with types of lymphoma…”. The text should be double checked for any other similar instances.

Reviewers' comments:

Reviewer's Responses to Questions

**Comments to the Author**

1. Is the manuscript technically sound, and do the data support the conclusions?

Reviewer #1: Yes

Reviewer #2: Yes

2. Has the statistical analysis been performed appropriately and rigorously? 

Reviewer #1: Yes

Reviewer #2: Yes

3. Have the authors made all data underlying the findings in their manuscript fully available?

Reviewer #1: Yes

Reviewer #2: Yes

4. Is the manuscript presented in an intelligible fashion and written in standard English?

Reviewer #1: Yes

Reviewer #2: Yes

5. Review Comments to the Author

Reviewer #1: Overall, this was a very interesting paper to review. The authors aimed to assess and map factors that influence haematologists HIV testing behaviour, and the results and conclusions align with this.

Areas that would benefit from further detail or discussion:

- The 5-point Likert scale that was used - how was scale ranked?

- Were questionnaire responses anonymous?

- The interviews ranged from 9-30 minutes – the manuscript would benefit from a discussion on this.

- It was written that eligible patients were given the opportunity to out of the use of their data - how was this done?

- The limited generalisability of the interviews was discussed, however, interviews aren’t necessarily intended to be generalisable.

Reviewer #2: This is a very nicely designed and presented study that presents the results of an audit of HIV testing in PWML as recommended by guidelines.

A few minor suggestions for inclusion. In table one add HIV positivity rates.

Discussion - comparison of responses to survey and those hospitals with lowest test rates. would focus on this - comments on verifying cost-effectiveness not aim of study.

all presented barriers are provider barriers and should be described as such. patient-level barriers are barriers perceived by patients. in this care older age, discomfort etc are provider barriers and should be labelled as such.

6. PLOS authors have the option to publish the peer review history of their article (what does this mean?). If published, this will include your full peer review and any attached files.

Reviewer #1: No

Reviewer #2: No

---

## [Author Response · Author response to Decision Letter 0]

7 Dec 2022

An extensive, point-by-point response to reviewer and editor comments was attached as a file.

---

## [Editor Report · Decision Letter 1]

19 Dec 2022

Mapping hematologists’ HIV testing behavior among lymphoma patients – A mixed-methods study

PONE-D-22-26866R1

Dear Dr. Bogers,

We’re pleased to inform you that your manuscript has been judged scientifically suitable for publication and will be formally accepted for publication once it meets all outstanding technical requirements.

Kind regards,

Simon White

Academic Editor

PLOS ONE
---

## [Editor Report · Acceptance letter]

22 Dec 2022

PONE-D-22-26866R1 

Mapping hematologists’ HIV testing behavior among lymphoma patients – A mixed-methods study 

Dear Dr. Bogers:

I'm pleased to inform you that your manuscript has been deemed suitable for publication in PLOS ONE. Congratulations! Your manuscript is now with our production department. 

Kind regards, 

on behalf of

Dr. Simon White 

Academic Editor

PLOS ONE